# Veterinary consumption of highest priority critically important antimicrobials and various growth promoters based on import data in Pakistan

Muhammad Umair[1], Samuel Orubu[2,3], Muhammad Hamid Zaman[3,4], Veronika J. Wirtz[4]*, Mashkoor Mohsin[1]*

1 Institute of Microbiology, University of Agriculture, Faisalabad, Pakistan, 2 Institute for Health System Innovation & Policy, Boston University, Boston, MA, United States of America, 3 Department of Biomedical Engineering, College of Engineering, Boston University, Boston, MA, United States of America, 4 Department of Global Health, Boston University School of Public Health, Boston, MA, United States of America

* mashkoormohsin@uaf.edu.pk (MM); vwirtz@bu.edu (VJW)

**Data Availability Statement:** All relevant data are within the paper and Supporting Information files.

**Funding:** The author(s) received no specific funding for this work.

## Abstract

### Background

Antimicrobial resistance (AMR) is a global public health emergency driven by the indiscriminate use of antimicrobial agents in humans and animals. Antimicrobial consumption surveillance guides its containment efforts. In this study, we estimated, for the first time, veterinary consumption of Critically Important Antimicrobials with Highest Priority (CIA-HtP) for Pakistan.

### Methods

The study used an export/import database which provided imports data collected from the Pakistan Customs Authority. We investigated imports of 7 CIA-HtP and various poultry feed additives/growth promoters (FAs/GPs) identified from a survey of 10 poultry and dairy farms in Punjab province in Pakistan and a previously published study, over a three-year period of 2017–2019. Antimicrobial consumption was estimated in mg/kg of country's animal biomass.

### Findings

Imports, in tonnes, for these 7 CIA-HtP were for the years 2017–19: tylosin 240.84, enrofloxacin 235.14, colistin 219.73, tilmicosin 97.32, spiramycin 5.79, norfloxacin 5.55, ceftiofur 1.02 for a total 805.39 tonnes. The corresponding antimicrobial consumption was 10.05 mg/kg of animal biomass. The poultry FAs/GPs contained: zinc bacitracin, enramycin, bacitracin methylene disalicylate, tylosin, tiamulin, colistin, lincomycin, streptomycin, flavophospholipol, tilmicosin, and penicillin with a total antimicrobial chemical compound (ACC) import volume of 577.18 tonnes for the years 2017–2019; and an estimated consumption of 96.53 mg/kg of poultry biomass.

**Competing interests:** The authors have declared that no competing interests exist.

## Interpretation

These antimicrobials were a mix of macrolides, quinolones, polymyxins and cephalosporins, among which are some also on the Watch or Reserve list by the WHO, indicating the need for stewardship and to conserve essential antimicrobials to contain AMR. The finding that a yearly average of 192.39 tonnes of the ACC imported were FAs/GPs further highlight the need for stronger regulation and enforcement.

## 1. Introduction

Antimicrobial resistance is a global threat to human and animal health. In 2015, the World Health Organization (WHO) adapted a Global Action Plan (GAP) on antimicrobial resistance (AMR) containment to mitigate the AMR threat. One of the main aims of the GAP is to reduce, and optimize, the use of antimicrobials in human and veterinary medicine. Since 2018, the Food and Agriculture Organization (FAO) and the World Organisation for Animal Health (WOAH) joined in WHO as a Tripartite Alliance to mitigate the AMR crisis [1] which is working to assist countries in preparing and implementing their National Action Plans (NAPs) on AMR. The Inter-Agency Coordination Group (IACG) on AMR had pointed out that many Low- and Middle-Income Countries (LMICs) lack the capacity to collect and analyze antimicrobial use (AMU) data in humans and animals [2]. Pakistan, in 2017, drafted its NAP on AMR with the key goal to monitor and reduce veterinary antimicrobial consumption (AMC) as a part of the global and national strategy to tackle AMR [3].

Monitoring requires structures–integrated/established data collection systems–for the routine assessment of AMC. In the animal health sector, the European Surveillance of Veterinary Antimicrobial Consumption (ESVAC) has established standardized methods to calculate veterinary AMU based on sales data. However, sales data is often difficult to obtain in many countries as they have not–contrary to the European Union countries or other countries such as Thailand enacted legislations that require manufacturers to report their sales data to the regulation authorities [4, 5]. The WOAH has also established a global database on antimicrobial agents intended for use in animals and encourage its member countries to participate in reporting their annual AMC data. In this context, WOAH published different templates to collect national-level animal AMC data based on the amounts of antimicrobial classes used or sold for medical use and growth promotion, further stratifying by different animals and administration routes. According to the WOAH's guidelines, AMC data can be obtained from different levels such as import, manufacturing, sales, end-user/farm-level [6].

Pakistan did not participate in WOAH's annual report, partly due to a lack of AMU monitoring and surveillance system in veterinary settings. In 2019, Pakistan's livestock sector contributed 12% to the GDP, comprising 61% of all agricultural output [7]. Pakistan is expected to be a large consumer of antimicrobials in the animal health sector because of its large animal population and emerging intensive livestock farming. Three farm-level studies, two from poultry [8, 9] and one in the dairy sectors [10] in Pakistan have confirmed this assumption. These studies highlighted the excessive veterinary use of Critically Important Antimicrobials with Highest Priority (CIA-HtP) for human medicine such as quinolones, third- and fourth-generation cephalosporins and macrolides as characterized by the WHO [11] which is concerning due to the increased risk of accelerating AMR. It has been extrapolated from these studies, that Pakistan is one of the high users of antimicrobials in food animals. In Pakistan, pharmaceutical importers, manufacturers, wholesalers, and veterinary professionals are not legally required to report

volumes of veterinary antimicrobial products imported, manufactured, sold, or prescribed. According to industry reports, there is low local manufacturing of pharmaceutical raw material (PRM), with almost 95% of the PRM is imported in Pakistan to be processed for product manufacturing by the human and veterinary pharmaceutical manufacturing industry [12, 13].

Thus, in the absence of more robust data collection tools, import volume can be used to estimating AMC at the national level to facilitate AMU monitoring. Translating AMU in animals into a standard metric for comparison requires adjustment for animal biomass. However, animal biomass in Pakistan has not been calculated.

This study, following up on the prior mentioned farm-level reports, aimed to assess the import volumes from 2017 to 2019 of CIA-HtP as well as various feed additives/growth promoters (FAs/GPs), in Pakistan. Veterinary AMC comprises both the use of the pure drug–the antimicrobial agent as a product in its own right for the control, prevention, or treatment of animal diseases–and the use of feed additives for growth promotion. The pure drug is imported either as a PRM, or active pharmaceutical ingredient (API), as well as a finished pharmaceutical product (FPP)–same as with growth promoters. Imports of the pure drug as a PRM is the cheaper alternative for Pakistan with a lot of veterinary manufacturing companies, and, thus, constitute the bulk of these imports. The scope of the present study is to analyze the volume of selected CIA-HtP PRMs and FAs/GPs imported in Pakistan between the years 2017 and 2019. Furthermore, import volumes were adjusted for animals biomass in Pakistan to calculate usage in mg per kg of animal biomass to estimate the national-level consumption of these antimicrobials and feed additives in the veterinary sector from 2017 to 2019.

## 2. Methodology

### Ethical statement

This work was granted exemption by the Institutional Bioethics/Biosafety Committee of the University of Agriculture, Faisalabad, Pakistan.

### Sample selection

Antimicrobial sample selection focused on the CIA-HtP identified from the data we had collected from an ongoing AMU surveillance study at ten dairy and broiler farms, five each, located in Punjab province for the months Sep-Oct 2020. At these farms, antimicrobial treatment data were collected, using a structured questionnaire, as tentative diagnosis, antimicrobial products, and quantities administered. Correspondents, that is, veterinarians, farm managers, or supervisors were instructed to keep records of all the antimicrobial products administered by taking product photographs using their smartphones, that were later shared through a social messaging application. Questionnaires were received as photographs or images (digital/soft copies) or as hard form (filled-in paper copies) via postal mail.

Antimicrobial chemical compounds (ACC) were identified from the product photographs. Any missing product was searched online, or the correspondents were contacted regarding any information required. The ACCs were then categorized according to the WHO list of Critically Important Antimicrobials (CIA) for human medicine [11]. All the seven listed CIA-HtP (ceftiofur, colistin, enrofloxacin, norfloxacin, spiramycin, tilmicosin, and tylosin) were selected for our study (Table 1).

### Data source and data collection

We used the Pakistan Exim Trade Info database as our import data source. This commercially available database contained all the imports of various goods in Pakistan and was accessible

**Table 1. Critically important antimicrobials with highest priority (CIA-HtP) identified at surveyed dairy and broiler farms.**

| Antimicrobial Class | Antimicrobial Active Ingredient |
| --- | --- |
| Cephalosporins 3rd | Ceftiofur |
| Polymyxins | Colistin |
| Quinolones and fluoroquinolones | Enrofloxacin |
| Quinolones and fluoroquinolones | Norfloxacin |
| Macrolides and ketolides | Spiramycin |
| Macrolides and ketolides | Tilmicosin |
| Macrolides and ketolides | Tylosin |

until July 2021 [14]. The data was made available from the publicly available international shipment records by the Federal Board of Revenue, Government of Pakistan. Pakistan import data is based on custom documents including "the bill of lading" to be filed with the customs authorities (Pakistan Customs Act 1969) [15]. The bill of lading contains consignment details including the date of shipment, importer/exporter and item description, gross weight, vessel name and route, agent, etc. We searched for all the seven CIA-HtP that were imported from January 1, 2017 to December 31, 2019. The data were downloaded as Microsoft Excel worksheets. Data for each antimicrobial import/shipment is captured as a single row with 15 variables entered in separate columns including the item description and shipment gross weight (Supplementary Metadata). Information on each shipment can be categorized into three groups i) importer and exporter details, ii) details on import commodities, and iii) details on the shipment date, identity, and shipping route. PRMs and FAs/GPs exporting countries were identified based on the exporter/manufacturer details and not on the vessel departure port which may mislead to the transit countries. The exporting countries identification was carried out by online searching the exporting companies' countries of origin.

These imports comprised several items, including PRMs or APIs for secondary manufacturing in Pakistan as well as the finished pharmaceutical products (FPP). We selected only PRM/API shipments as these constitute the bulk or is the major marker of consumption of veterinary antimicrobials in Pakistan.

For the PRMs registered for use in both the human and veterinary medicine, veterinary imports were selected by identifying the importers manufacturing veterinary FPPs from the Drug Regulatory Authority of Pakistan (DRAP) minutes of meetings [16] and by online search of the importers/manufacturers websites.

AMC estimation was performed by summation in tonnes, following net weight determination, of all the imported PRMs and analysis of veterinary FAs/GPs followed by adjustment for animals biomass.

## Import volumes calculation

There were three separate net weight calculations for PRMs. Using shipments with PRMs net weights (NW) available in the item description column, we calculated a percent packaging weight (PPW), using NW and gross weight (GW), for each PRM (Eq 1). This PPW was applied to all other shipments without stated net weights for "single-item" (or shipments with one PRM imported) to calculate a PRM adjusted net wight (AdjNW) by subtracting the PPW from each shipment's gross weight (Eq 2). For shipments with more than one PRM/item imported "multiple-item shipments" and no net weights available in item description, gross weights were distributed equally over the number of items in that shipment (for example, for the multiple-item shipment of two products enrofloxacin HCl and oxytetracycline HCl with a gross

weight of 609 kg, 304.5 kg was considered for the calculation of enrofloxacin HCl net weight). The AdjNW for multiple-item shipments were calculated similarly as for the single-item shipments with no net weights available in item description (Eq 2). Thus, there were two net weights the i.e., NW and AdjNW. A total net weight (TNW) for each PRM was then calculated by adding these two net weights (Eq 3).

$$PPW = \left[ \frac{\sum_{m=1}^{M} (NW)_m}{\sum_{m=1}^{M} (GW)_m} \times 100 \right] - 100 \tag{1}$$

$$PRM_{AdjNW} = \sum_{n=1}^{N} (GW - PPW)_n \tag{2}$$

$$PRM_{TNW} = \left[ \sum_{m=1}^{M} (NW)_m \right] + PRM_{AdjNW} \tag{3}$$

Where M is the total number of shipments with PRM net weight available, and N is the total number remaining shipments during (2017–19).

All the net weight calculations for the 7 different CIA-HtPs were made against their chemical compounds given in S3, S4 Tables in S1 File [6]

## Veterinary feed additives/growth promoters data collection and analysis

Veterinary FAs/GPs are finished pharmaceutical products ready to be administered in feed. FA/GP products were identified from the previously published data [8] and PRM shipments' item description where multiple items were imported. Identified products were searched on Pak Exim Trade info and the shipments with more than one item imported were examined to look for other FA/GP products. Net weights for these products were calculated from the packaging information obtained from the manufacturer or importer website (Eq 4). For the products with no information available, net weights were calculated as per the methodology described for PRMs. The amounts of ACC/s for each feed FA/GP product were calculated from product strengths and volumes imported (Eq 5).

$$FA/GP_{NW} = Package\ NW \times Tot.\ Packges\ Imported \tag{4}$$

$$ACC = \%\ Strength \times FA/GP_{NW} \tag{5}$$

Where package NW is the net weight of FA/GP in each package. The % strength is the grams of API in each 100 grams of the FA/GP product.

It is important to note that these FAs/GPs were exclusively imported to be used in poultry feed as evident from the importers that were poultry feed manufacturers or their suppliers. Pakistan's total poultry counts show 95% (1279.76/1353.24 Million no.) of the broiler population, major consumer of FA/GP products, compared to 4% (59.82/1353.24 Million no.) of the layers in 2019–20 [7].

## Animal biomass calculation

Animal biomass for the species likely to be treated with antimicrobial agents was calculated as per the WOAH methodology [6]. Data on live animals and livestock primary (producing animals/slaughtered, yield, and production quantity) for the years 2017–19 were obtained from online database of the Statistics Division of Food and Agriculture Organization of the United

Nations (FAOSTAT) [17, 18]. Buffaloes, cattle, goats, sheep, chickens, camels, asses, mules, and horses were identified for the calculation of animals biomass. Live weights for camels, asses, mules, and horses were calculated from the regional livestock primary data (Asia) [18] as their livestock primary data were incomplete or not available for Pakistan. For buffaloes and cattle, population proportions (P.pop) for calves (<1 years), young (1–3 years), and adult (>3 years) were calculated from Pakistan Livestock Census 2006 [19]. Calves, young, and adults values for P.pop for buffaloes were found 29.36%, 11.48%, and 59.16%, respectively, whereas for cattle these values were 24.82%, 9.87%, and 65.31%, respectively. Calculation for the total biomass of different animal species for the years 2017–19 are detailed in the supplementary material (S2 Table and Eq S1-S4 in S1 File). PRMs were adjusted for the total animal biomass whereas the FA/GP ACCs were adjusted for the total chicken biomass.

The adjusted consumption $total_{mg/kg}$ for the years 2017–19 (y = 17–19) was calculated using Eq 6.

$$Total_{mg/kg} = \frac{\sum_{y=17}^{19} PRM \ or \ ACC}{\sum_{y=17}^{19} Animal \ Biomass} \tag{6}$$

## 3. Results

### Veterinary CIA-HtP pharmaceutical raw materials import

A total of 1,607 PRM shipments between the years 2017 and 2019 were examined for the seven CIA-HtP PRMs. Import volumes of 1,359 sea- and 91 air-, or 90.2% (1,359/1,607), shipments were considered for the calculation of net weights for CIA-HtP PRMs. The rest 9.2% (147/1,607) of the shipments were human imports, finished pharmaceutical products, or FAs/GPs, and thus excluded (Supplementary Metadata). During the three years a total of 805.39 tonnes, with an annual average of 268.46 tonnes, of CIA-HtP PRMs were imported with all the shipments originating from China except for six sea- and two air-shipments originating from India and one air-shipment from UK (Table 2 and Fig 1). Highest import volume was observed during the year 2019 with a total import of volume of 288.83 tonnes followed by 2017 (284.63 tonnes) and 2018 (231.93 tonnes) (Table 2 and Fig 2). Tylosin (240.84 tonnes), enrofloxacin (235.14 tonnes), and colistin (219.73 tonnes) were the top three antimicrobials imported during the three years followed by tilmicosin (97.32 tonnes), spiramycin (5.79 tonnes), norfloxacin (5.55 tonnes), and ceftiofur (1.02 tonnes) with an average annual import volumes of: 80.28, 78.38, 73.24, 32.44, 1.93, 1.85, and 0.34 tonnes, respectively. (Table 2).

**Table 2. Import volumes of WHO critically important antimicrobials with highest priority (CIA-HtP) for the years 2017–19.**

| Antimicrobial | PRM Import volumes (tonnes) | | | | mg/kg (total animal biomass) | | | |
|---|---|---|---|---|---|---|---|---|
| | 2017 | 2018 | 2019 | Total | 2017 | 2018 | 2019 | Total |
| Tylosin | 88.38 | 66.41 | 86.05 | 240.84 | 3.42 | 2.49 | 3.11 | 3 |
| Enrofloxacin | 80.8 | 78.01 | 76.33 | 235.14 | 3.13 | 2.92 | 2.76 | 2.93 |
| Colistin | 91.78 | 54.55 | 73.4 | 219.73 | 3.55 | 2.04 | 2.66 | 2.74 |
| Tilmicosin | 18.52 | 29.17 | 49.63 | 97.32 | 0.72 | 1.09 | 1.8 | 1.21 |
| Spiramycin | 2.41 | 1.61 | 1.77 | 5.79 | 0.09 | 0.06 | 0.06 | 0.07 |
| Norfloxacin | 2.4 | 1.66 | 1.49 | 5.55 | 0.09 | 0.06 | 0.05 | 0.07 |
| Ceftiofur | 0.34 | 0.52 | 0.16 | 1.02 | 0.01 | 0.02 | 0.01 | 0.01 |
| Total | 284.63 | 231.93 | 288.83 | 805.39 | 11.02 | 8.68 | 10.45 | 10.05 |

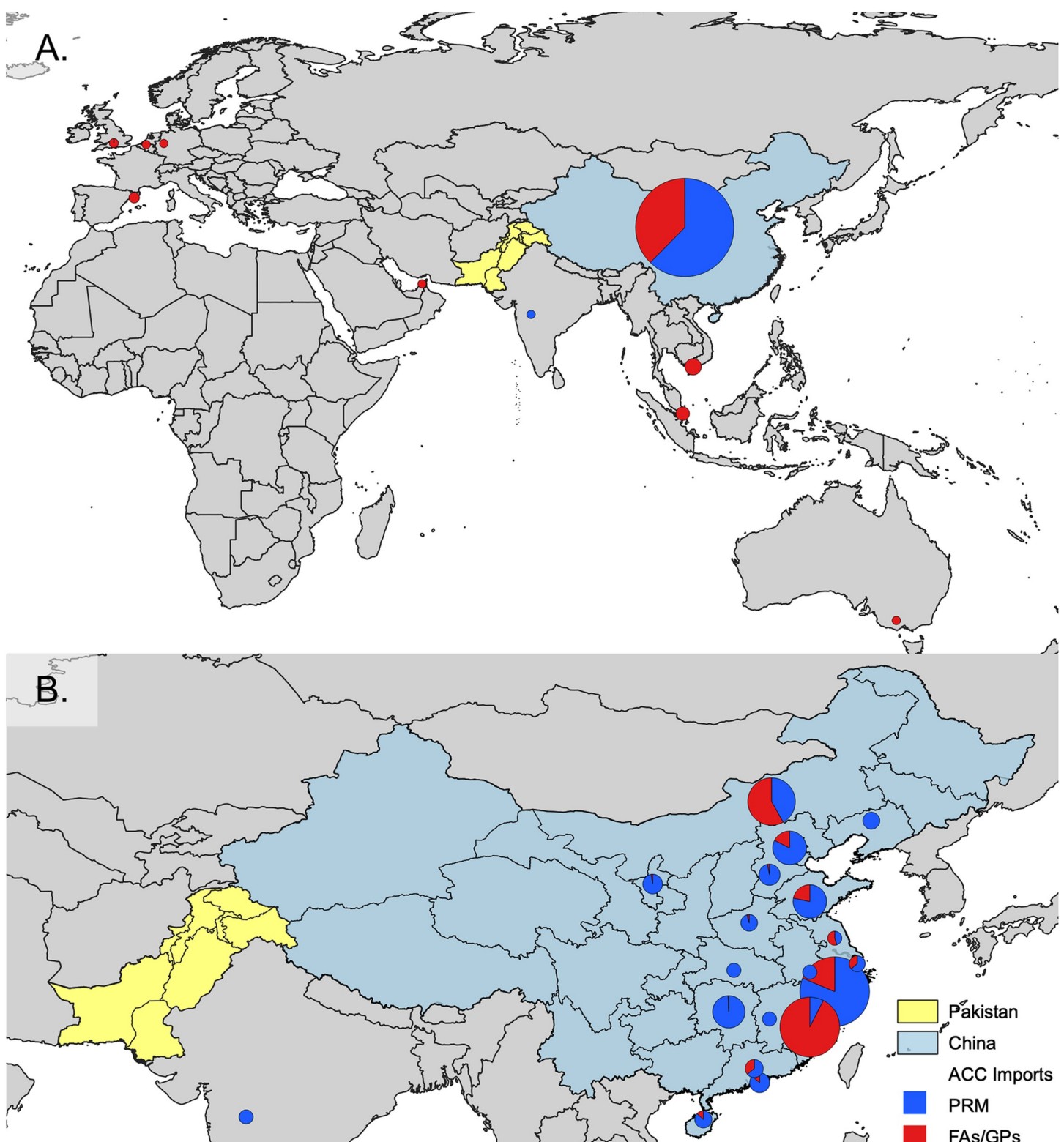

**Fig 1.** Total volumes and ratios of pharmaceutical raw materials and feed additives/growth promoters antimicrobial chemical compounds for the years 2017–19 imported from: A. different countries and B. China and its different provinces.

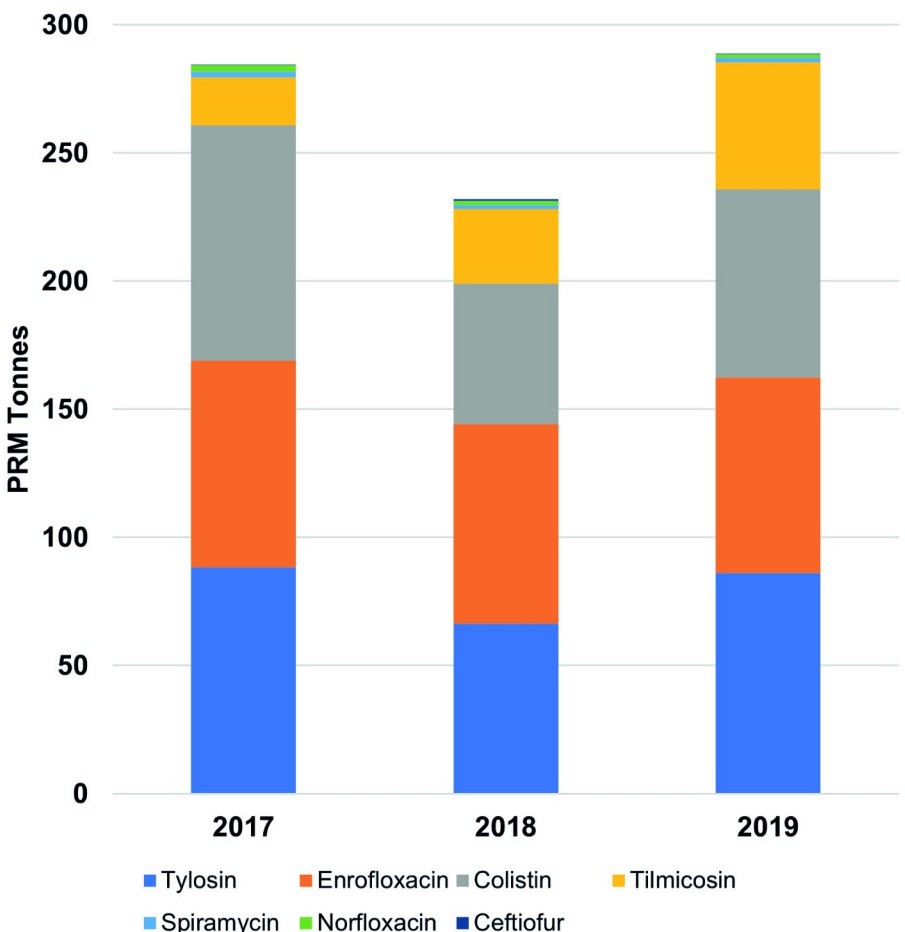

**Fig 2. Import volumes of WHO critically important antimicrobials with highest priority (CIA-HtP) pharmaceutical raw materials identified at surveyed farms for the years 2017–19.** For small amounts imported referred to Table 2.

## Veterinary feed additives/growth promoters import

A total of 22 products and 21 FA/GP ACCs strengths were identified in 383 shipments imported from China (289/383), Singapore (32/383), Vietnam (32/383), UK (17/383), Spain (6/383), Australia (3/383), Belgium (3/383), and Germany (1/383). These FAs/GPs were exclusively imported to be used in poultry feed as evident from the importers that were poultry feed manufacturers or their suppliers (S1 Fig in S1 File).

In terms of the imports by ACCs during the three years a total of 577.18 tonnes of FA/GP ACCs were imported with an annual average of 192.39 tonnes. Highest ACC import volume was observed for the year 2017 (233.33 tonnes), followed by 2019 (201.89 tonnes) and 2018 (141.96 tonnes). Zinc bacitracin (316.4 tonnes), enramycin (62.58 tonnes) and bacitracin methylene disalicylate (55.29 tonnes) were the top three FA/GP ACCs followed by tylosin (40.69 tonnes), tiamulin (32.51 tonnes), colistin (20.2 tonnes), lincomycin (19.08 tonnes), streptomycin (11.88 tonnes), flavophospholipol (9.59 tonnes), timlicosin (5 tonnes), and penicillin (3.96 tonnes) with annual average volumes of 105.47, 20.86, 18.43, 13.56, 10.84, 6.73, 6.36, 3.96, 3.2, 1.67, and 1.32 tonnes, respectively (Table 3 and Fig 3). The total FA/GP products volume which is ACC/s plus excipients weight can be found in the supplementary material (S1 Table and S1, S2 Figs in S1 File).

**Table 3. Antimicrobial chemical compounds import volumes for poultry feed additives/growth promoters for the years 2017–19.**

| Antimicrobial | ACC Import volumes (tonnes) | | | | mg/kg (chicken biomass) | | | |
|---|---|---|---|---|---|---|---|---|
| | 2017 | 2018 | 2019 | Total | 2017 | 2018 | 2019 | Total |
| Zinc Bacitracin | 159.83 | 56.7 | 99.87 | 316.4 | 87.68 | 28.58 | 45.97 | 52.92 |
| Enramycin | 15.02 | 30.28 | 17.28 | 62.58 | 8.24 | 15.26 | 7.95 | 10.47 |
| BMD | 7.49 | 15.05 | 32.75 | 55.29 | 4.11 | 7.59 | 15.07 | 9.25 |
| Tylosin | 18.85 | 9.85 | 11.99 | 40.69 | 10.34 | 4.96 | 5.52 | 6.81 |
| Tiamulin | 12.32 | 12.83 | 7.36 | 32.51 | 6.76 | 6.47 | 3.39 | 5.44 |
| Colistin | 7.5 | 3.11 | 9.59 | 20.2 | 4.11 | 1.57 | 4.41 | 3.38 |
| Lincomycin | 6.61 | 5.1 | 7.37 | 19.08 | 3.63 | 2.57 | 3.39 | 3.19 |
| Streptomycin | 4.28 | 4.18 | 3.42 | 11.88 | 2.35 | 2.11 | 1.57 | 1.99 |
| Flavophospholipol | 0 | 3.47 | 6.12 | 9.59 | 0 | 1.75 | 2.82 | 1.6 |
| Tilmicosin | 0 | 0 | 5 | 5 | 0 | 0 | 2.32 | 0.84 |
| Penicillin | 1.43 | 1.39 | 1.14 | 3.96 | 0.78 | 0.7 | 0.52 | 0.66 |
| Total | 233.33 | 141.96 | 201.89 | 577.18 | 128 | 71.56 | 92.93 | 96.53 |

BMD: Bacitracin Methylene Disalicylate

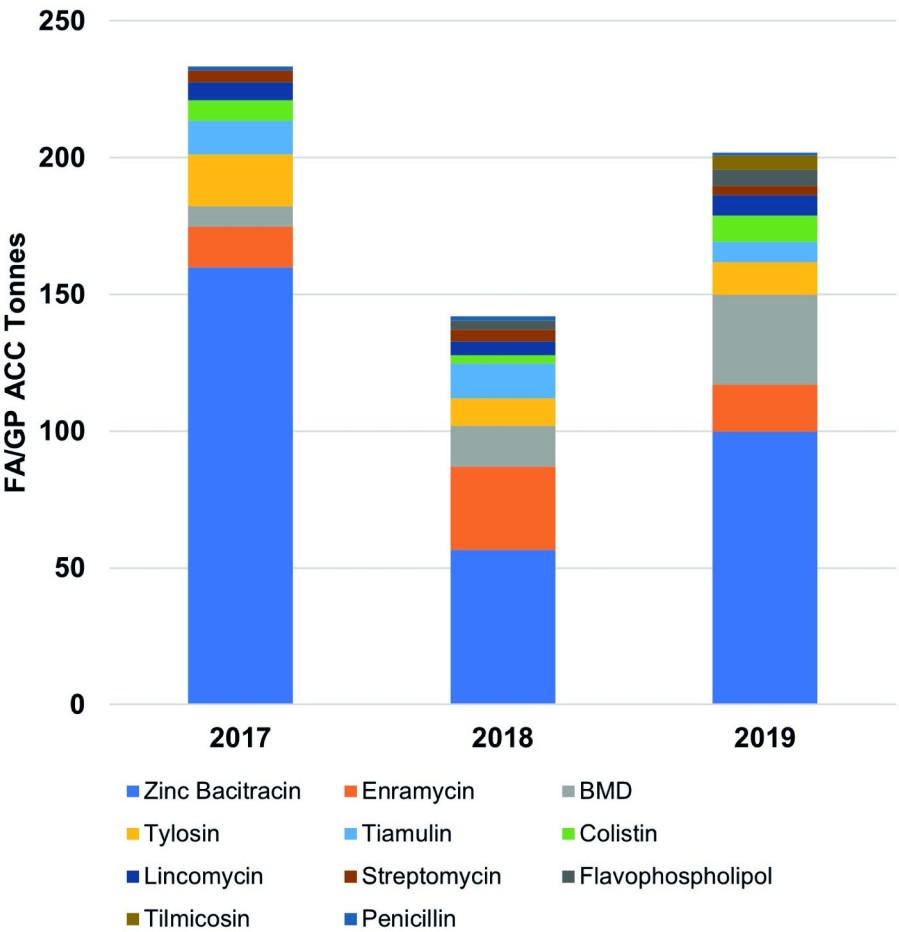

**Fig 3. Import volumes of veterinary feed additive/growth promoter antimicrobial chemical compounds for the years 2017–19.** For small amounts imported referred to Table 3.

## Animal biomass

Total animal biomass for the species likely to be treated with antimicrobial agents for the years 2017, 2018, and 2019 was 27638.5, 26713.11, and 25822.73 thousand tonnes (T tonnes), respectively. Highest biomass, with average values for three years, was calculated for cattle (8855.66 T tonnes), followed by buffaloes (7829.63 T tonnes), goats (4536.98 T tonnes), sheep (2063.29 T tonnes), chickens (1993.12 T tonnes), asses (911.83 T tonnes), camels (388.8 T tonnes), horses (100.97 T tonnes), and mules (44.5 T tonnes). Average total animal biomass for the years 2017–19 was calculated to be 26724.78 T tonnes (S2 Table in S1 File and Fig 4).

## CIA-HtP and FAs/GPs consumption

When adjusted for total animals biomass for the years 2017–19 we found tylosin (3 mg/kg), enrofloxacin (2.93 mg/kg), and colistin (2.74 mg/kg) were the top three CIA-HtP PRMs consumed followed by tilmicosin (1.21 mg/kg), spiramycin and norfloxacin (0.07 mg/kg each), and ceftiofur (0.01 mg/kg). The highest total consumption was recorded for the year 2017 (11.02 mg/kg) followed by 2019 (10.45 mg/kg) whereas lowest was observed for the year 2018 (8.68 mg/kg). Total PRM consumption for the years 2017–19 was 10.05 mg/kg of the cumulative animals biomass (Table 2 and Fig 5).

Adjusted for total chicken biomass zinc bacitracin (52.92 mg/kg), enramycin (10.47 mg/kg), and bacitracin methylene disalicylate (BMD) (9.25 mg/kg) were the top three feed FA/GP ACCs consumed followed by tylosin (6.81 mg/kg), tiamulin (5.44 mg/kg), colistin (3.38 mg/kg) lincomycin (3.19 mg/kg), streptomycin (1.99 mg/kg), flavophospholipol (1.6 mg/kg), tilmicosin (0.84 mg/kg), and penicillin (0.66 mg/kg). Highest FA/GP ACCs consumption was recorded for the year 2017 (128 mg/kg) followed by 2019 (92.93 mg/kg) whereas lowest

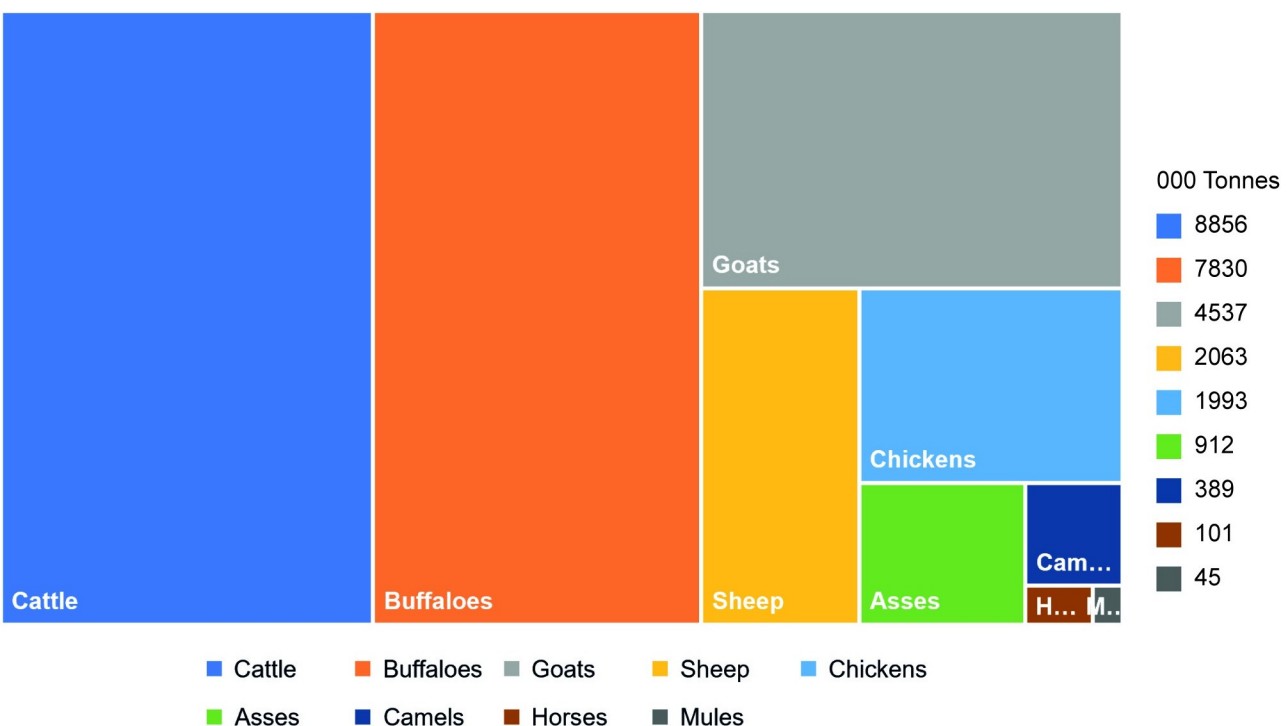

**Fig 4. Average biomass of different animal species likely to be treated with antimicrobials for the years 2017–19.**

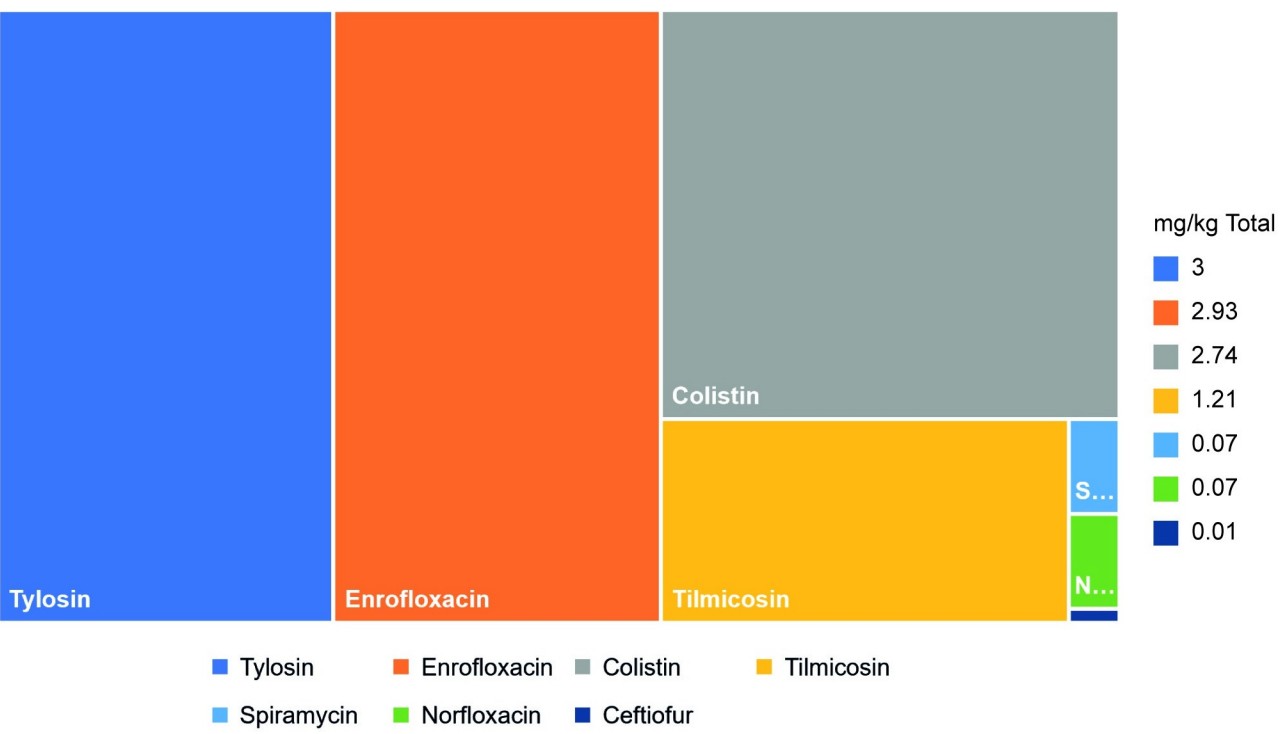

mg/kg Total
- 3
- 2.93
- 2.74
- 1.21
- 0.07
- 0.07
- 0.01

■ Tylosin   ■ Enrofloxacin   ■ Colistin   ■ Tilmicosin
■ Spiramycin   ■ Norfloxacin   ■ Ceftiofur

**Fig 5. Total import volumes of pharmaceutical raw materials adjusted for total animal biomass for the years 2017–19.**

consumption was observed for 2018 (71.56 mg/kg). Total consumption for the years 2017–19 was 96.53 mg/kg of the cumulative chicken biomass (Table 3 and Fig 6).

## 4. Discussion

AMC monitoring is an essential step in implementing reduction targets in national efforts to contain AMR. The global consumption of antimicrobials in animals is on rise due to rapidly growing intensive livestock farming operations partially driven by the increased human demand of animal source proteins. This is more noticeable in LMICs with the lack of regulation and monitoring systems for use of antimicrobials in animal health sector [20]. However, some middle-income countries have made significant progress by implementing effective legislation requiring manufacturers to report antimicrobials sales volume such as Thailand which has enabled the country to report its national consumption [21].

Antimicrobials used in food-producing animals is closely related to those used in human medicine and can select for resistant bacteria in human and animals. The FAO-WOAH-WHO Tripartite Alliance on AMR urges countries to monitor the extent of antimicrobial consumption in all sectors including animal husbandry and to share this information so that the global consumption trends can be monitored [22]. Advances in collective knowledge of the dynamics between antimicrobial use and resistance has promoted the need for antimicrobial surveillance, with this information applied to policies and practices aimed at AMR reduction. For example, the European Union, leaders in recognition of the link between use and resistance, have long-running surveillance systems to inform both policy and practice. The European Surveillance for Antimicrobials Consumption (ESVAC) reported an overall decrease in the sales of veterinary antimicrobials in 31 European countries from 2010 to 2018. The decrease was significantly impacted by the decline in sales of antimicrobials with high risk to public health

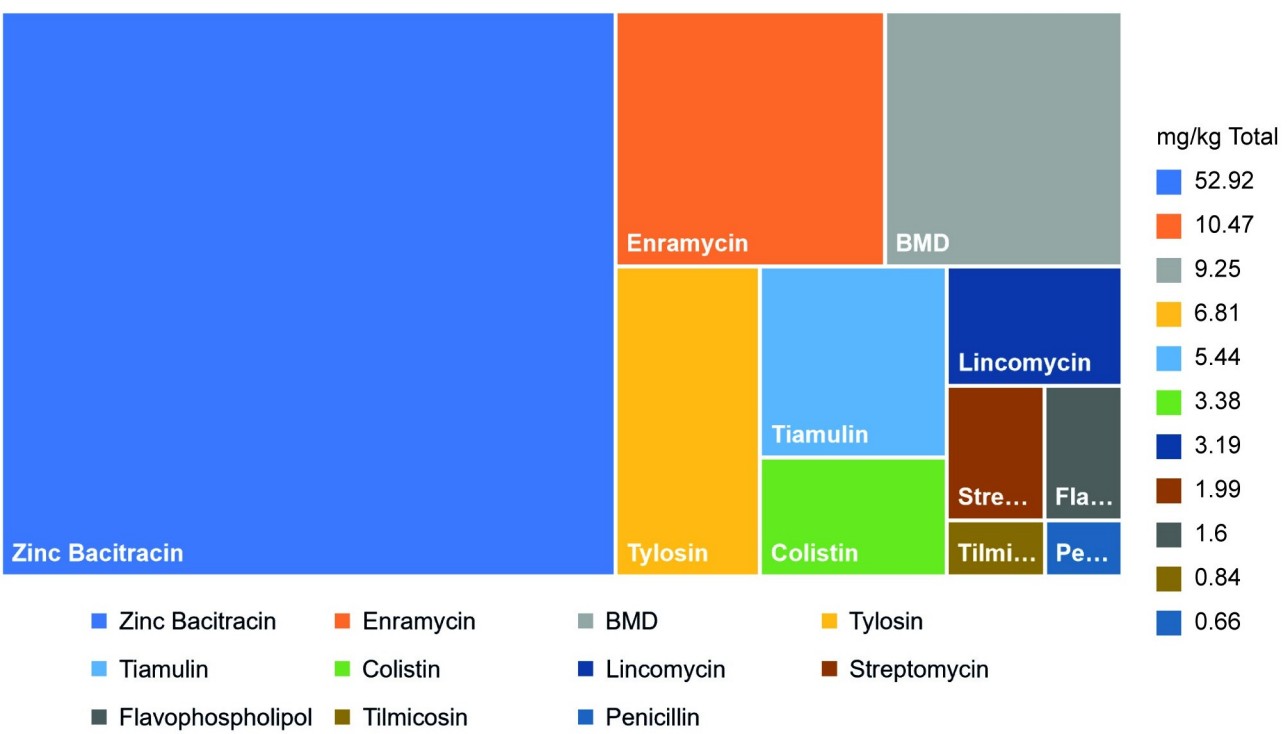

**Fig 6. Total import volumes of feed additive/growth promoter antimicrobial chemical compounds adjusted for total chicken biomass for the years 2017–19.**

[4]. A similar decreasing trend was reported from other high-income countries (HIC) and some upper-middle income countries such as China, which is attributed to effective AMU stewardship efforts [23]. Thus, routine collection, and use, of AMU and AMC data, as these examples suggest, is an important tool to promote public health, with regards to AMR containment.

Contrastingly to findings from HICs where the use of medically important antimicrobials in food animals are declining, we found in Pakistan no decrease in the import of these antimicrobials along with the growth promoters during the study period 2017–2019.

This is one of the first studies providing a longitudinal evaluation of national-level antimicrobial consumption in the veterinary sector for Pakistan. The study demonstrated that approximately 805 tonnes of seven selected CIA-HtP (net weight of PRM) were imported between 2017 and 2019, an average of 268 tonnes per year. Although this volume far exceeds the total yearly import volumes of veterinary antimicrobials in other LMICs i.e., Cameroon (36 tonnes) [24] and Timor-Leste (57 kg) [25], these differences could be due to size of animal population and farming types. Antimicrobial growth promoters (AGPs) are banned in many countries and others are in process to phase out the use of AGPs. Our study fills an important knowledge gap by describing import quantities, consumption, and details on exporting countries of FAs/GPs in the veterinary sector. For Pakistan, during the three years studied, 577 tonnes of ACCs were imported as FAs/GPs, this indicates the FAs/GPs are still being used in food animals production despite their ban in countries from Asia [26, 27] and wider [28, 29]. Thus, it establishes a baseline for AMU in the veterinary sector in Pakistan.

Overall, this study contributes to our understanding of AMC in animals in Pakistan in a variety of ways. First, it demonstrates that import data can be used to estimate the consumption of CIA-HtP in animals and unregulated antimicrobial feed additives in Pakistan. Import

data are particularly relevant as other data sources such as manufacturers, wholesalers, retailers, and farmers data are challenging to obtain in Pakistan as in many other countries. Ideally, nationally representative data should be collected at farm level which is the point of usage. Although a series of AMU studies have been carried out on broiler [8, 9] and dairy farms [10] in Pakistan but the samples are not nationally representative and hence cannot be used to estimate the national veterinary AMC. Pakistan does not require wholesalers or distributors to report their sales data to the DRAP. This is different from Europe and Thailand, as mentioned above, where many countries require wholesalers to report on antimicrobial sales data [4, 9]. It is imperative to understand the amount of CIA-HtP and more broadly CIAs used in animals to formulate regulations and stewardships programs for optimal use.

Second, our study provides an estimate on the high-level consumption of CIAs with highest priority. These antimicrobials belong to macrolides, quinolones, polymyxins and cephalosporins listed in the Watch and Reserve categories of WHO AWaRE classification indicating antimicrobials with higher resistance potential and reserved for multi-drug resistant infections [30]. Our results are in line with other studies on animal farms which reported that tylosin, enrofloxacin and colistin are one of the most commonly used CIAs in animals [8]. Colistin is an important last resort antimicrobial in the treatment of multi-drug resistance infections in human. However, colistin resistance due to plasmid-mediated mobile colistin resistance *mcr-1* gene is emerging in animals, human, and environment in Pakistan [31–33], and such a high level import could poses a serious threat to the emergence of colistin resistant bacteria in human clinical settings. Colistin being the third largest import volume of all CIA-HtP is concerning and raises important questions for policy makers in the urgency to regulate its use in the veterinary sector in Pakistan.

Third, our finding that yearly average of 192 tonnes of the ACCs imported were FAs/GPs highlights the need for better regulation and enforcement that goes beyond the nominal discussion on prescription, but also includes the use of antimicrobials for growth promotion, an issue that has not gotten any serious attention in Pakistan [34]. It is also noteworthy that these feed additives are for consumption exclusively in broiler as growth promoters considering the fact that chicken meat consumption is much higher than other types of meat consumption in Pakistan [35].

Antimicrobials as growth promoters were banned in the European Union countries in 2006 and later in UK, USA, and some other countries including Thailand, Indonesia, and Vietnam resulting in subsequent lower AMC especially for colistin [26]. Here, we found colistin based FA/GP products import estimating approximately 175 tonnes per year (524.2 tonnes during 2017–19) (S1 Table in S1 File). It also means that surveillance of antimicrobial consumption in Pakistan must include feed additives given their importance. Unfortunately, imports of FA/GP antimicrobials are not being regulated by DRAP, therefore their imports and subsequent use remains unchecked. Formulation and implementation of policies restricting the consumption of colistin in agriculture could have a significant effect on reducing colistin resistance in both animals and humans as reported from China [36].

Finally, our study is the first that estimates Pakistan's animal biomass and the consumption of antimicrobials per unit animals biomass. Veterinary AMC studies based on import data are scarce. We found tylosin (3 mg/kg), enrofloxacin (2.93 mg/kg), and colistin (2.74 mg/kg) among the top three CIA-HtP (PRM) used in veterinary sector in Pakistan. When considering import volumes and subsequent use of growth promoters in chicken zinc bacitracin (52.92 mg/kg), enramycin (10.47 mg/kg), and BMD (9.25 mg/kg) ranked among the top FA/GP ACCs. To our knowledge, this is the first study to calculate antimicrobial growth promoter consumption adjusted for animal biomass. However, it is difficult to compare the veterinary AMC per unit biomass due to lack of any standardized metric and monitoring system [37].

Moreover, our findings are difficult to be compared with other AMU studies because we focused only on critically important antimicrobials and not all classes of antimicrobials.

## Limitations

Here we note the limitations of our study which are relevant to consider when discussing our findings. Import data may overestimate consumption as some imported antimicrobials may expire before manufactured, sold, or consumed. Ideally, antimicrobial consumption is measured through observation at farm level. However, this is costly and nationally representative data are currently unavailable. Our study also did not comprehensively measure all antimicrobials and focused exclusively on the CIA-HtP that were used on ten broiler and dairy farms in two provinces in Pakistan. There may be more critically important antimicrobials used on Pakistani farms; however, based on the present survey and previously published studies [8–10] we reasonably assume that the seven CIA-HtP we included in our study are the most frequently used ones. Furthermore, we were unable to ensure that we did not miss any feed additives that contain antimicrobials as we do not have a comprehensive list of all feed additives sold in Pakistan. The calculations were based on PRM or the ACC of active ingredient (moiety) however, WOAH accepts AMC reporting based on the amount of ACC [6]. Moreover, the antimicrobial products exported by Pakistani manufacturers were not adjusted in our consumption calculations, reason being the lack of exports data availability on product names and manufacturers. We assume that Pakistan imports of finished veterinary antimicrobial products are generally low due to their prices multiple time higher than their local alternatives and limited market. Chemicals, drugs, and medicine are among the top imports by Pakistan whereas their exports are ranked 21st of 24 products as reported in the Pakistan Economic Survey 2019–20 [7].

## 5. Conclusion

Pakistan's veterinary pharmaceutical industry is importing large volumes of critically important antimicrobials with highest priority for human medicine. These antimicrobials are being consumed predominantly in food animals including poultry and dairy. Moreover, the imports of exceedingly high volumes of antimicrobials as feed additives/growth promoters, exclusively used in poultry, remains unregulated. Unfortunately, at the moment there is no legislation in any house of parliament or any government authority on surveillance of the imports and consumption of these products. The lack of awareness, oversight, or regulation has resulted in imports of additives that include critically important antimicrobials. With a high population burden and weak AMR surveillance, continued oversight of imports that include life-saving antimicrobials as feed additives are likely to make the problem of AMR much worse and could further undermine the already strained and underfunded health system in Pakistan.

The lack of legislation and surveillance on the imports and consumption of these products in veterinary sector could expose the population of 208 million on the verge of AMR crisis. Strict and careful national and international laws and strategies will be required to check the production and exports of growth promoters and critically important antimicrobials to avoid any potential global health crisis.

## Supporting information

**S1 File.**
(PDF)

**S2 File. Pharmaceutical raw materials and feed additives/growth promoters shipments and calculations metadata included in this study.**
(ZIP)

## Author Contributions

**Conceptualization:** Muhammad Umair, Veronika J. Wirtz, Mashkoor Mohsin.

**Data curation:** Muhammad Umair.

**Formal analysis:** Muhammad Umair.

**Investigation:** Muhammad Umair, Mashkoor Mohsin.

**Methodology:** Muhammad Umair, Samuel Orubu, Veronika J. Wirtz, Mashkoor Mohsin.

**Software:** Muhammad Umair.

**Supervision:** Muhammad Hamid Zaman, Veronika J. Wirtz, Mashkoor Mohsin.

**Validation:** Samuel Orubu, Veronika J. Wirtz.

**Visualization:** Muhammad Umair.

**Writing – original draft:** Muhammad Umair, Samuel Orubu, Veronika J. Wirtz, Mashkoor Mohsin.

**Writing – review & editing:** Muhammad Umair, Samuel Orubu, Muhammad Hamid Zaman, Veronika J. Wirtz, Mashkoor Mohsin.

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
