## [Decision Letter · Decision Letter 0]

17 Mar 2022

PONE-D-21-39799Veterinary Consumption of Highest Priority Critically Important Antimicrobials and Various Growth Promoters based on Import Data in PakistanPLOS ONE

Dear Dr. Mohsin,

Thank you for submitting your manuscript to PLOS ONE. After careful consideration, we feel that it has merit but does not fully meet PLOS ONE’s publication criteria as it currently stands. Therefore, we invite you to submit a revised version of the manuscript that addresses the points raised during the review process.

I found the manuscript interesting as a good start for these kind of studies. Although the reviewers have indicated several loopholes in study design and interpretation, I will be interested to know how the authors address the issues. Please submit your revised manuscript by May 01 2022 11:59PM. If you will need more time than this to complete your revisions, please reply to this message or contact the journal office at plosone@plos.org. Please include the following items when submitting your revised manuscript:A rebuttal letter that responds to each point raised by the academic editor and reviewer(s). You should upload this letter as a separate file labeled 'Response to Reviewers'.A marked-up copy of your manuscript that highlights changes made to the original version. You should upload this as a separate file labeled 'Revised Manuscript with Track Changes'.An unmarked version of your revised paper without tracked changes. You should upload this as a separate file labeled 'Manuscript'.

We look forward to receiving your revised manuscript.

Kind regards,

Indranil Samanta

Academic Editor

PLOS ONE

Journal Requirements:

3. We note that Figures 1 and S1 in your submission contain map images which may be copyrighted. All PLOS content is published under the Creative Commons Attribution License (CC BY 4.0), which means that the manuscript, images, and Supporting Information files will be freely available online, and any third party is permitted to access, download, copy, distribute, and use these materials in any way, even commercially, with proper attribution. For these reasons, we cannot publish previously copyrighted maps or satellite images created using proprietary data, such as Google software (Google Maps, Street View, and Earth). For more information, see our copyright guidelines: http://journals.plos.org/plosone/s/licenses-and-copyright.

a. You may seek permission from the original copyright holder of Figures 1 and S1 to publish the content specifically under the CC BY 4.0 license.  

5. Please upload a copy of Supporting Information table S3 and S4 which you refer to in your text on page 8.

Additional Editor Comments (if provided):

I found the manuscript interesting as a good start for these kind of studies. Although the reviewers have indicated several loopholes in study design and interpretation, I will be interested to know how the authors address the issues.

Reviewers' comments:

Reviewer's Responses to Questions

**Comments to the Author**

1. Is the manuscript technically sound, and do the data support the conclusions?

Reviewer #1: Yes

Reviewer #2: No

2. Has the statistical analysis been performed appropriately and rigorously? 

Reviewer #1: Yes

Reviewer #2: I Don't Know

3. Have the authors made all data underlying the findings in their manuscript fully available?

Reviewer #1: Yes

Reviewer #2: Yes

4. Is the manuscript presented in an intelligible fashion and written in standard English?

Reviewer #1: Yes

Reviewer #2: Yes

5. Review Comments to the Author

Reviewer #1: Summary of research

The authors estimate the quantity of selected antimicrobials that are imported into Pakistan. The selected antimicrobials include highest priority critically important antimicrobials (CIA-HtP) that were being used on 10 poultry and diary farms; and poultry feed additives/growth promoters identified from a previous study. The quantity of CIA-HtP was estimated in tonnes of pharmaceutical raw material (PRM) and adjusted for total animal biomass. The quantity of poultry feed additives/growth promoters was estimated in tonnes of active pharmaceutical ingredient (AAI) and adjusted for total chicken biomass. Using this data, the paper identifies the top CIA-HtP and feed additives/growth promoters that are imported into Pakistan. It states that the use of antibiotics into Pakistan is high compared to other countries, although it is hard to make direct comparisons due to differing methodologies in calculation. It cautions that stronger regulation and enforcement is needed in Pakistan to mitigate the risk of antimicrobial resistance (AMR).

Overall impression

This was a decent attempt at quantifying antimicrobial imports in Pakistan focusing on selected CIA-HtPs and feed additives/growth promoters. The Methods and Result section are well written and transparent although small sections of the methodology require amendments to improve clarity. Equations 1 to 6 are outside of the peer-reviewer’s expertise to comment.

The Discussion and Conclusion section requires substantial development and redrafting before publication. There are several sweeping and unsubstantiated statements which requires referencing (e.g. line 237-238) or elaboration (e.g. line 274-275). The study also needs a deeper technical interpretation of the findings and the findings need to be linked to other relevant research, even if it is outside of Pakistan. Several key points can be better articulated and developed (e.g. Line 190-192; Line 260-268).

Finally, there are grammatical errors that need to be addressed to improve readability, especially in the Discussion section.

Discussion of specific areas for improvement (Major issues)

• Line 99 and Line 300-301: It should be clarified why CIA-HtPs were only identified on dairy and broiler farms but not on other types of farms. This is especially important since other species of animals (e.g. buffalos, goats and sheep) contribute to a significant portion of total animal biomass. See Figure 4 in manuscript. In line 301, explain to the audience why it is a reasonable assumption that the 7 CIA-HtPs were the most frequent ones used on farms.

• Line 173: It should be explained why only total chicken biomass was used. Perhaps total broiler biomass should be used instead, since the growth promoters and feed additives were only used on broilers.

• Line 190-192: An effort could be made to discuss the origin of antimicrobials into Pakistan in the Discussion section. For example, if these exporting locations were the country of manufacture or transit.

• Line 237-238: This statement requires substantiation with a reference. Studies that have shown a decline (instead of increase) in the use of medically important antimicrobials in food animals should also be considered.

• Line240: This should be PRM. See Table 2 and line 181.

• Line 245: Suggestion removal of the word “companies”, as the main manuscript does not include any description on companies exporting growth promoters in the veterinary sector.

• Line 260-268: It seems that the main point of this paragraph is that there is a high level of use of CIAs in animals in Pakistan. This was not clear from the first sentence in line 260. In addition, line 161 and 162 could be further developed in terms of comparing Pakistan’s level of CIA use in animals to other countries with a similar agriculture system.

• Line 272: This statement regarding the lack of serious attention in Pakistan on the need for better regulation and enforcement requires substantiation or a reference.

• Line 272-273: The point that feed additives were used exclusively in broilers should be mentioned in the “Results” section.

• Line 274:275: There is a leap of logic in this statement. High use of antimicrobials not mean that there will be high residue levels in the meat, as long as withdrawal periods are adhered to. Residues in milk can also be discussed since it is a large industry in Pakistan.

• Line 279: It is unclear where 175 tonnes per year was obtained from as this value was not mentioned in the manuscript. Also clarify if this numbers refers to PRM of the 7 CIA-HtP, AAI of chicken feed additive/growth promoters or both.

• “Limitations” section (line 294-309): The inclusion of only PRMs (and hence exclusion of active pharmaceutical ingredient and finished pharmaceutical products) for the 7 CIA-HtPs is a limitation of this study – this was not mentioned. Separately, the authors could comment on the impact of using PRM to estimate import volume, and why they were not converted to AAI.

• The Abstract should not contain information that was not mentioned in the main manuscript (e.g. Watch or Reserve list by WHO)

Discussion of specific areas for improvement (Minor issues)

• Grammar issues (e.g. Line 34, 244, 245-247, line 288, line 306-307)

• Line 58: OIE has introduced a method to collect national-level consumption data which adjusts for biomass. The template does not adjust for biomass.

• Line 73: Please clarify if there are very few companies manufacturing AAI, or a very low volume of AAIs being manufactured by companies.

• Line 79: Normalization is a statistical term which has not been used in this field. OIE uses the term adjusted. (i.e. adjusted for animal biomass). This applies to the rest of the manuscript which uses the term “normalization”.

• Line 104: Please clarify the statement “questionnaires were recollected as photographs…”. Recollect might not be a appropriate term here.

• Line 243: The authors could be more specific here. Is farming more intensive in Pakistan?

• Line 284: Consider the use of “consumption” instead of “exposure”

In summary, many issues in this paper can be addressed with major revisions and the authors should strive for publication of this paper.

Reviewer #2: Antimicrobial resistance is a global health crisis which results from extensive use of antimicrobial drugs (AM) in many sectors, including agriculture. Reduction and improvement in use in animal agriculture is necessary on a global scale, but requires documentation of current “baseline” usage in different countries. For many countries it is hard to obtain this data. This paper uses 2017-2019 Pakistan customs importation data to assess food animal usage to obtain data that would otherwise be unobtainable and to calculate national consumption data, since all raw AMs, finished products or their active ingredients are imported into the country. It converts this data into a standard “animal biomass” metric, which allows comparison to usage in other countries.

It is an interesting paper that documents importation of AMs into Pakistan, and to some extent their likely use. However there are considerable weaknesses in the data.

The strengths of the paper are its description of the quantities of HPCIAs imported into Pakistan, as well as of growth promoters (GPs) and feed additives over time, and the use of import data for this purpose. Although “To best of our knowledge, this is the first to describe quantities, consumptions, and details on exporting countries/companies of growth promotors in veterinary sector”, does this really matter. It’s a good start and deserves to have this element published.

The demonstrated that approximately 805 tonnes of seven selected CIA-HtP (net weight of 240 AAI) were imported between 2017 and 2019, an average of 268 tonnes per year. Although the authors seem surprised that “this volume far exceeds the total yearly import volumes of veterinary antimicrobials in other lower- and middle-income countries” actually they are probably not surprising. India used 10,000 tons of colistin in animal feed annually until recently.

The weaknesses of the paper are numerous, and not summarized in the Limitations section.

For example, different assumptions were made of the relative weight content of shipments of 7 High Priority Critically Important Antimicrobials (HPCIA) described as more than one (or multiple) AM, or where indeed no net weight was given. For AM growth promoters, all imported as finished products, estimates were made of the amount of active ingredient imported and net weights also calculated.

The number and biomass of a wide variety of animals regarded as likely to be treated with AMs (buffaloes, cattle, goats, sheep, chickens, camels, asses, mules, and horses) and slaughtered for meat consumption were calculated from FAO data, and live weights estimated for these livestock from Pakistan or, if not available, from Asian generic sources. Pharmaceutical raw ingredients usage was calculated from total animal biomass and growth promoter active ingredient use by chicken biomass. Pakistan does not grow swine so that it is assumed probably reasonably that all growth promoters go to chickens. However, we really do not know where these AMs went and whether these animals are treated (or fed) with AMs, so that Figures 5 and 6 may be meaningless.

The authors described sampling on 10 dairy and 10 chicken farms, but never refer to this data again; it would have been useful, perhaps. In the absence of knowing how these AMs are used, the data are not helpful, especially adjusted to “biomass” metrics. There is no attempt to compare such metrics with other countries, which would have been interesting.

The different metrics used are confusing: for example, “805 tonnes of seven selected CIA-HtP (net weight of 240 AAI) were imported”. Why use two different metrics?

Minor comments

Figure 1: These are confusing since it says “different countries” but only shows China. I now see that the figure does show countries and indicates the relative contribution by the size of the circle; most are almost invisible. I’d delete the confusing Figure and just keep the data in the text. Figure 1b (not labelled as such) seems to show different Provinces in China, why not say this in the legend, and if this is correct does this matter? What is the difference from Figure S1?

Figure 2: It’s hard to see the colours for norfloxacin and ceftiofur, in fact I don’t see any ceftiofur. Make a note in the legend (Table 2 shows a very small amount imported). What’s the difference in Figure 3 and Figure S2?

The Figure legend for Figure 4 is very poor; there are no units given. Do you mean relative?

Table S2 should be given in the main paper, it’s important.

Zinc bacitracin and BMD are the same drug essentially, bacitracin.

“During the three years a staggering 577 tonnes of antimicrobial active ingredients were imported as growth promoters, which is more than two thirds of the total CIA-HtP imported of this study”. However, the great majority of GPs are not CIA-Htp (HPCIAs), so it’s not clear what this statement means.

6. PLOS authors have the option to publish the peer review history of their article (what does this mean?). If published, this will include your full peer review and any attached files.

Reviewer #1: **Yes: **Shawn Ting

Reviewer #2: No

---

## [Author Response · Author response to Decision Letter 0]

9 Jun 2022

Dear Solna C Santos,

PLOSOne Editorial Office

We confirmed that all figures for this manuscript, including Figure 1 and Figure S1 have been created by the authors. None of these figures have been copied from other sources including not from https://pak.eximtradeinfo.com So the copyright is with the authors and not pak.eximtradeinform.com. 

With this information, we ask the editor to please proceed and accept all figures and tables.

Dr. Mashkoor Mohsin

Corresponding author

---

## [Editor Report · Decision Letter 1]

17 Aug 2022

Veterinary Consumption of Highest Priority Critically Important Antimicrobials and Various Growth Promoters based on Import Data in Pakistan

PONE-D-21-39799R1

Dear Dr. Mohsin,

We’re pleased to inform you that your manuscript has been judged scientifically suitable for publication and will be formally accepted for publication once it meets all outstanding technical requirements.

Kind regards,

Indranil Samanta

Academic Editor

PLOS ONE

Additional Editor Comments (optional):

I am pleased to notice that the authors have provided sufficient justifications of their findings based on the reviewer comments and the manuscript is also improved a lot. It is a good start for the studies related to quantification of antibiotics used in Veterinary sector from LMICs and it's consequences in context of antimicrobial resistance.
---

## [Editor Report · Acceptance letter]

22 Aug 2022

PONE-D-21-39799R1 

Veterinary Consumption of Highest Priority Critically Important Antimicrobials and Various Growth Promoters based on Import Data in Pakistan 

Dear Dr. Mohsin:

I'm pleased to inform you that your manuscript has been deemed suitable for publication in PLOS ONE. Congratulations! Your manuscript is now with our production department. 

Kind regards, 

on behalf of

Dr. Indranil Samanta 

Academic Editor

PLOS ONE